# Training DNNs with Hybrid Block Floating Point

**Mario Drumond**
Ecocloud
EPFL
mario.drumond@epfl.ch

**Tao Lin**
Ecocloud
EPFL
tao.lin@epfl.ch

**Martin Jaggi**
Ecocloud
EPFL
martin.jaggi@epfl.ch

**Babak Falsafi**
Ecocloud
EPFL
babak.falsafi@epfl.ch

## Abstract

The wide adoption of DNNs has given birth to unrelenting computing requirements, forcing datacenter operators to adopt domain-specific accelerators to train them. These accelerators typically employ densely packed full-precision floating-point arithmetic to maximize performance per area. Ongoing research efforts seek to further increase that performance density by replacing floating-point with fixed-point arithmetic. However, a significant roadblock for these attempts has been fixed point's narrow dynamic range, which is insufficient for DNN training convergence. We identify block floating point (BFP) as a promising alternative representation since it exhibits wide dynamic range and enables the majority of DNN operations to be performed with fixed-point logic. Unfortunately, BFP alone introduces several limitations that preclude its direct applicability. In this work, we introduce HBFP, a hybrid BFP-FP approach, which performs all dot products in BFP and other operations in floating point. HBFP delivers the best of both worlds: the high accuracy of floating point at the superior hardware density of fixed point. For a wide variety of models, we show that HBFP matches floating point's accuracy while enabling hardware implementations that deliver up to $8.5\times$ higher throughput.

## 1 Introduction

Today's online services are ubiquitous, offering custom-tailored content to billions of daily users. Service customization is often provided using deep neural networks (DNNs) deployed at a massive scale in datacenters. Delivering faster DNN inference and more accurate training is often limited by the arithmetic density of the underlying hardware platform. Most service providers resort to GPUs as the platform of choice for training neural networks because they offer higher arithmetic density per silicon area than CPUs, through full precision floating-point (FP32) units. However, the computational power required by DNNs has been increasing so quickly that even traditional GPUs cannot satisfy the demand, pushing both accelerators and high-end GPUs towards narrow arithmetic to improve logic density. For instance, NVIDIA's Volta [1] GPU employs half-precision floating point (FP16) arithmetic while Google employs a custom 16-bit floating point representation in the second and third versions of the TPU [2] architecture.

Following the same approach, there have been research efforts to replace narrow floating point with even denser fixed-point representations. Fixed-point arithmetic promises excellent gains in both speed and density. For instance, 8-bit fixed-point multipliers occupy $5.8\times$ less area and consume $5.5\times$ less energy than their FP16 counterpart [3]. Unfortunately, training with fixed-point, or even with FP16, has yielded mixed results due to the limited range inherent in these representations [4].

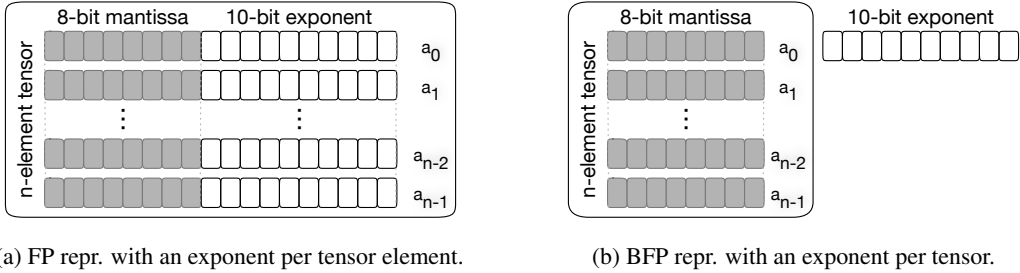

(a) FP repr. with an exponent per tensor element.    (b) BFP repr. with an exponent per tensor.

Figure 1: A $n$-element tensor in BFP and FP representations. BFP tensors save space and simplify computations by sharing exponents across tensors.

Block floating point (BFP) is an alternative representation that strikes a balance between logic density and representable range. Signal processing platforms have historically resorted to BFP to optimize for both performance and density, but BFP has not been thoroughly investigated in the context of DNN training. Figure 1 highlights the difference between the BFP and FP representations. Floating point encodes numbers with one exponent per value (Figure 1a), requiring complex hardware structures to manage mantissa alignment and exponent values. In contrast, BFP (Figure 1b) shares exponents across blocks of numbers (or tensors), which enables dense fixed point logic for multiply-and-accumulate operations. In the past, signal processors leveraged BFP to convert common algorithms (e.g., FFT) to fixed point arithmetic hardware. DNN computations, like signal processing, consist mostly of MAC-based operations (i.e., dot products) and therefore can benefit from BFP's arithmetic density.

While promising, replacing floating point with BFP for DNN training faces three significant challenges. First, although BFP dot products are very area-efficient, other BFP operations may not be as efficient, leading to hardware with floating-point-like arithmetic density. Second, exponent sharing may lead to data loss if exponent values are too large or too small, making the exponent selection policy a crucial design choice in BFP-based systems. Finally, BFP may incur data loss if the tensors' value distributions are too wide to be captured by its mantissa bits.

In this paper, we target the three aforementioned challenges with a hybrid approach, performing all dot products in BFP and other operations in floating point. First, we observe that, since dot products are prevalent in DNNs, performing other operations with floating point incurs low area overhead, similar to the overhead incurred by arbitrary BFP operations.

Solving the second problem of selecting the right exponents is equivalent to selecting good scales for quantization points. Prior work [5, 6] introduced coarse-grained exponent selection algorithms. These algorithms train DNNs with fixed point and adjust the exponents a few times per training batch or epoch. Unfortunately, the coarse-grained approaches fail to accommodate drastic exponent changes, resulting in data loss and hurting convergence. One solution to that problem is to use wider mantissas paired with conservatively large exponents so that there is some headroom in case tensor values are unexpectedly large. These strategies result in less dense hardware due to the wide arithmetic used and also introduce more hyperparameters, further complicating the training process. We argue that a more aggressive approach to exponent selection, with exponents chosen before each dot product takes place, leads to convergence on narrower mantissas.

Finally, we target the third challenge by only converting values to BFP right before dot products, leveraging the fact that dot products are resilient to the input data loss incurred by BFP. Other operations take floating points as inputs, enabling the accurate representation of arbitrary value distributions.

**This paper's contributions are:** (1) a hybrid BFP-FP (HBFP) DNN training framework that maximizes fixed-point arithmetic and minimizes the mantissa width requirements while preserving convergence, (2) two optimizations to BFP, namely tiling and wide weight storage, to improve BFP's precision with modest area and memory bandwidth overhead, (3) an exploration of the HBFP design space showing that DNNs trained on BFP with 12- and 8-bit mantissas match FP32 accuracy, serving as a drop-in replacement for this representation and (4) we show, with an FPGA prototype, that HBFP exhibits arithmetic density similar to that of fixed-point hardware with the accuracy of FP32 hardware.

## 2   Related Work

Training and inference in DNNs with narrow representations are well studied subjects. In this section, we review prior work.

**Hybrid accelerators.**   The separation between dot products and other operations already exists in commodity hardware in NVIDIA Volta's FP16 Tensor Cores [1] and in Google's Tensor Processing Unit [7] architecture. We take one step further and use different numeric representations for these different operations, enabling training with dense fixed-point arithmetic.

**Inference with reduced precision.**   Quantization [8] is a widely used technique for DNN inference. BFP [9] has also been proposed for inference. These techniques quantize the weights of DNNs trained with full precision floating point to use fixed-point logic during inference. We consider the more challenging task of training DNNs with arithmetic density that matches quantized inference.

**Binarized and ternary neural networks.**   Binarized [10] and Ternary [11, 12] neural networks are another way to compress models. Although these networks enable inference with hardware that is orders of magnitude more efficient than floating-point hardware, they are trained like traditional neural networks, with both activations and parameters represented with floating point. Therefore, these approaches are orthogonal to BFP-based training. Other work [13, 14] uses binary operations for forward and backward passes but not for weight gradient calculation and accumulation. The new training algorithm is not transparent to users, requiring redesign of networks with numeric representation in mind. In contrast, our approach is backwards compatible with FP32 models.

**Training with end-to-end low precision.**   ZipML [15], DoReFa [6], and Flexpoint [5] train DNNs with end-to-end low precision. They use fixed-point arithmetic to represent weights and activations during forward and backward passes, and introduce various algorithms and restrictions to control the numeric range of activations or select quantization points for the fixed-point representations.

DoReFa [6] requires techniques to control the activations' magnitudes, and is unable to quantize the first and last layers of networks. Others [15, 16] take a more theoretical approach to find the optimal quantization points for each dataset, performing both computations and communication using fixed-point arithmetic. We use BFP instead, effectively computing quantization points by choosing exponents at a finer granularity, before every dot product.

Flexpoint [5] performs all computations in fixed point. It uses the Autoflex algorithm twice per minibatch to predict the occurrence of overflows and adjust the tensor exponents accordingly. They leverage the slowly changing aspect of gradients exponents to minimize the number of exponent updates. However, to minimize overflows, they end up requiring conservatively large exponents, leaving the higher bits of mantissas unused and increasing mantissa width. Furthermore, Autoflex adds an artificial dependency between computations when it collects tensor value stats, making it unsuitable for DNNs that employ dynamic dataflow and limiting training scalability since it restricts the way DNNs can be sliced for distributed training. Our approach computes exponents more frequently and it does so in-device, without requiring any additional stat collection, and accommodating dynamic dataflows naturally. We observe that, as long as dot product calculation's intermediate values remain in fixed-point-like representations, conversions are infrequent enough that the hardware area dedicated to them accounts for an insignificant fraction of the total accelerator area.

## 3   Specialized Arithmetic for DNNs

Due to the massive computational requirements for DNNs employed in datacenter-scale online services, operators such as Google have started adopting specialized numeric representations for DNNs. So far, accelerators have employed fixed-point representations for inference [7], and narrow floating-point representations [1, 17] for training. From a hardware design perspective, the use of reduced-precision arithmetic allows silicon designers to improve logic density and energy efficiency, while minimizing the number of bits used to represent models relaxes demands on both memory capacity and bandwidth. From the user's perspective, arithmetic representations must be easy to use, without sacrificing accuracy or requiring any algorithmic techniques to recover performance.

Table 1: Validation test error of ResNet-20 on CIFAR-10 with narrow FP representations.

| Mantissa bit-widths | | | | | Exponent bit-widths | | |
|---|---|---|---|---|---|---|---|
| 2 | 4 | 8 | 24 | | 2 | 6 | 8 |
| N/A | 9.77% | 8.05% | 8.42% | | N/A | 14.67% | 8.42% |

FP32 representations are easy to use but inefficient. They represent numbers with a 24-bit mantissa and a 8-bit exponent. In terms of precision, the 24-bit mantissa is an overkill for DNNs. Table 1 shows the validation error obtained when training ResNet-20 models on CIFAR10 using floating-point representations with various mantissas and exponent widths. We observed convergence without loss of precision with 8-bit mantissas, convergence with a small loss of precision with 4-bit mantissas, and divergence only when using 2-bit mantissas. Exponent width, however, cannot be reduced because of its impact over numeric range. We observed diminished validation precision when reducing the exponent width from 8 to 6 bits, and divergence when using 2-bit exponents.

Hardware developers have made the same observation, leading the state of the art to quickly drift towards narrow floating-point representations. One prominent example is FP16. FP16 is denser than FP32, employing 11-bit mantissas and 5-bit exponents. However, FP16's logic overhead is still high compared to that of fixed point. For instance, although the area of an FP16 multiplier is $4.7\times$ smaller than that of a FP32 multiplier, it is still $5.8\times$ larger than its 8-bit fixed-point counterpart [3]. FP16 also introduces complexity for users, as the 5-bit exponent results in a narrow range that is not sufficient to represent gradients throughout the training process [4]. DNN training requires numeric representations with wide range because, as the loss value and the learning rates decrease, the gradient values also decrease, often by several orders of magnitude. To mitigate this issue, Google has moved to a 16-bit floating point [2] representation that employs 8 bits for both the mantissas and the exponents, to improve the dynamic range.

Given these requirements, we identify block floating point (BFP) as the ideal numeric representation for DNNs. Like floating point, BFP represents numbers with mantissas and exponent and therefore exhibits a wide dynamic range. However, BFP logic is denser because exponents are shared across entire tensors, resulting in dot products that can be computed entirely in fixed-point logic. Because the vast majority of the arithmetic operations executed by DNN training and inference are dot products, we are able to fold almost all the training computation into fixed-point logic.

## 4  DNN Training With BFP Arithmetic

Equation (1) computes the real value $a_i$ of an element $i$ of a BFP tensor $a$ with mantissa $m_i^a$ and exponent $e_a$.

$$a_i = m_i^a \times 2^{e_a} \tag{1}$$

In this example, BFP can only represent $a$ accurately if the value distribution of $a$ is not too wide to be captured by $m^a$ and the exponent $e_a$ is representative of said value distribution. If $e_a$ is too large then small values are lost and the most significant bits of the mantissas are wasted. If $e_a$ is too small, then the larger values in $a$ will be saturated, leading to data loss.

Equation (2) calculates the dot product between BFP tensors $a$ and $b$, each with $N$ elements.

$$a \cdot b = \sum_{i=1}^{N} \left( (m_i^a \times 2^{e_a}) \times (m_i^b \times 2^{e_b}) \right) = 2^{e_a + e_b} \times (m^a \cdot m^b) \tag{2}$$

The dot product $m^a \cdot m^b$ is computed entirely in fixed-point arithmetic, without the alignment of intermediate values, since all elements $m_i^a$ and $m_i^b$ are fixed point. In a matrix multiplication $A \times B$, it is enough for $A$ to have one exponent per row, and $B$ to have one exponent per column. BFP matrix multiplications can also be tiled. With tiled matrices, tile multiplications are performed in fixed point, and their results are accumulated in floating point arithmetic, requiring mantissa realignment.

### 4.1  Hybrid Block Floating Point (HBFP) DNN Training

We propose the use of BFP for all dot product computations, with other operations performed in floating-point representations. This configuration enables the bulk of the DNN operations to be

performed in fixed-point logic and facilitates the use of various activation functions or techniques like batch normalization without the restrictions imposed by BFP.

HBFP is superior to pure BFP for two reasons. First, using BFP for all operations may lead to divergence unless wide mantissas are employed. DNN operations often result in tensors with wide value distributions, that can be too wide for BFP, leading to loss of values at the edge of the distributions (i.e., values that are too small or too large). Thus, most operations cannot tolerate taking BFP values as inputs, as they may change the value distributions in non-trivial ways, with both small and large values having an impact on the results. Dot products, however, do not face this problem. Dot products are reductions, and thus, the input tensors' largest values dominate the sum, with small values having little impact on the final result. Consequently, BFP dot products tolerate data loss as long as tensor exponents are large enough to avoid saturation.

The second reason is the area overhead of general purpose BFP operations. BFP can lead to costly floating-point-like hardware in the general case, since it may lead to numerous mantissa realignments and expensive exponent computations to ensure that tensors' exponents match value distributions. BFP dot products are denser because the overhead of exponent calculations and mantissa realignments is amortized over the reduction. For instance, in a dot product between two tensors with $N$ elements, BFP leads to one mantissa realignment for every $2 \times N$ operations, while a BFP ReLU operation, for instance, requires one mantissa realignment per operation.

We propose to use BFP in all dot-product-based operations present in DNNs (i.e., convolutions, matrix multiplications, and outer products), and floating-point representations for all other operations (i.e., activations, regularizations, etc). We store long-lasting model state (i.e., weights) in BFP and transient activation values in floating point. We convert tensors to BFP before every dot product, using the exponent of the largest tensor value, and convert the result back to floating point afterwards.

### 4.2 Minimizing BFP Data Loss

The amount of data loss incurred by BFP is determined by two factors: the size of tensors that share exponents and the width of the mantissas. We devise two optimizations to mitigate each of these issues with modest silicon area and memory bandwidth overhead: tiling and wide weight storage.

**Tiling:** Matrix multiplications are often tiled to improve locality in intermediate caching storage. We observe that BFP can also benefit from tiling. More specifically, we divide the weight matrices in tiles of a predefined size and share exponents within tiles. Tiling bounds the number of values that share exponents, reducing data loss. This optimization incurs some silicon density penalty because the resulting tiles need to be accumulated using floating-point arithmetic. Nevertheless, the overhead is small: for a tile size of $N \times N$, tiling incurs one extra floating-point operation every $2 \times N$ operations. For large tiles, the area of the extra floating-point adders is negligible compared to the area of the $N$ multiply and accumulate units.

**Wide weight storage:** To minimize data loss in long-lasting training state, we store weights with wider mantissas. All operations are still executed using the original mantissa, and only weight updates use the wider mantissa. Therefore, we still reduce the memory bandwidth requirements for forward and backward passes, during which only the most significant bits of the weights are accessed. The least significant bits of the weight matrices are only accessed by weight updates.

## 5 Methodology

### 5.1 HBFP Simulation on GPU

We train DNNs with the proposed HBFP approach, using BFP in the compute-intensive operations (matrix multiplications, convolutions, and their backward passes) and FP32 in the other operations. We simulate BFP dot products in GPUs by modifying PyTorch's [18] linear and convolution layers to reproduce the behaviour of BFP matrix multipliers. We redefined PyTorch's convolution and linear modules using its *autograd.function* feature to create new modules that process the inputs and outputs of both the forward and backward passes to simulate BFP. In the forward pass, we convert the activations to BFP, giving the $x$ tensor one exponent per training input. Then we execute the

target operation in native floating-point arithmetic. In the backward pass, we perform the same pre-/post-processing of the inputs/outputs of the $x$ derivative.

We handle the weights in the optimizer. We created a shell optimizer that takes the original optimizer, performs its update function in FP32 and converts the weights to two BFP formats: one with wide and another with narrow mantissas. The former is used in future weight updates while the latter is used in forward and backward passes. We also use this same mechanism to simulate different tile sizes for weight matrices. Finally, for convolutional layers, we tile the two outer feature map dimensions of the weight matrices.

## 5.2 Evaluation Setup

**Datasets.** We experiment with a set of popular image classification tasks with the CIFAR-100 [19], SVHN [20], and ImageNet [21] datasets. We used standard data augmentation [22, 23] for CIFAR-100 and no augmentation for SVHN. We also evaluate language modeling tasks with Penn Tree Bank(PTB) dataset [24].

**Evaluation metrics.** To evaluate the impact of HBFP and explore the design space of different BFP implementations, we tune the models using FP32, and then train the same models from scratch with the same hyperparameters in HBFP. For the image classification experiments, we report training loss and validation top-1 error. For the language modeling models, we report training loss and validation perplexity.

**Training.** We use a WideResNet [25] trained on CIFAR-100 to explore the BFP design space, evaluating models trained with various mantissa widths and various tile sizes. To show that HBFP is a viable alternative to FP32, we train a wide range of models using various datasets. We train ResNet [22], WideResNet [25], and DenseNet [26] models on the CIFAR-100 and SVHN datasets; a ResNet model on ImageNet and the LSTM from [27] on PTB. We trained all models using the same hyperparameters reported in their respective original papers.

## 5.3 Hardware Prototype Implementation

HBFP accelerators exhibit arithmetic density that is similar to their fixed-point counterparts. To further illustrate this point, we synthesized a proof-of-concept FPGA-based accelerator. Figure 2 shows the block diagram of the accelerator. Grey boxes and arrows indicate buffers, units, and dataflow in BFP format while other colors correspond to FP. We implemented the basic operations needed for neural network training (i.e., matrix multiplication, transpose, convolutions, outer product, weight update, and data movement operations) using a dataflow similar to [28].

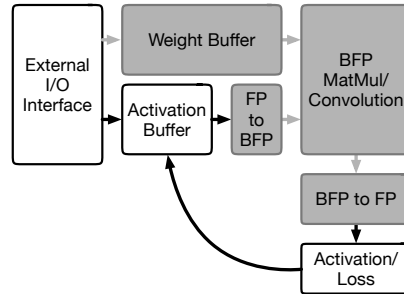

Figure 2: HBFP accelerator with BFP.

We employ a matrix multiplication (MatMul) unit followed by an activation/loss unit, sized to maximize resource utilization in the FPGA. The MatMul output width matches the activation/loss units' input width to avoid backpressure. The FP-to-BFP units detects the maximum exponent of incoming FP tensors and normalizes their mantissas accordingly, while the BFP-to-FP unit takes the results computed in the wide accumulators present in the MatMul unit, normalizes and truncates their mantissas, and computes their exponents. Hence, the MatMul unit never causes overflows or saturation. We employ stochastic rounding [29] during mantissa truncation, using a Xorshift random number generator [30]. Xorshift is a very small random number generator, employing three constant shifts and three xor operations, and it has been shown [31] to work well for stochastic rounding. Finally, weight updates are done entirely in the activation unit, in floating point. The proof-of-concept accelerator operates with both weights and activations stored on-chip.

Table 2: Test error of image classification models. RN, WRN and DN indicate ResNet, WideResNet and DenseNet, respectively. *hbfpX_Y* indicates an experiment with X-bit mantissas and Y-bit weight storage and a tile size of 24. All dot product operations are performed in X-bit arithmetic.

| | CIFAR 100 | | | SVHN | | | ImageNet |
|---|---|---|---|---|---|---|---|
| | RN-50 | WRN-28-10 | DN-40 | RN-50 | WRN-16-8 | DN-40 | RN-50 |
| fp32 | 26.07% | 20.35% | 26.03% | 1.89% | 2.00% | 1.80% | 23.64% |
| hbfp8_16 | 25.12% | 20.78% | 26.27% | 1.98% | 1.98% | 1.79% | 23.88% |
| hbfp12_16 | 25.10% | 20.78% | 25.82% | 1.96% | 1.94% | 1.85% | 23.58% |

# 6 Evaluation

We now evaluate DNN training with HBFP. We explore the design space of BFP, finding the best-performing configurations of BFP. We vary both the mantissa width and the tile sizes. Then, we move on to evaluate HBFP on various datasets and tasks, to show that HBFP is indeed a drop-in replacement for FP32. Finally, we evaluate the throughput gains obtained with HBFP using our hardware prototype.

**BFP design space:** We train WideResNet-28-10 models on CIFAR-100 using various HBFP configurations. To experiment with the mantissa width, we train models with 4-, 8-, 12- and 16-bit wide mantissas. All models with mantissas wider than 8 bits result in final validation error within 1% of the FP32 baseline, with only 4-bit mantissas showing a large accuracy gap, with 4.1% larger error. We also evaluate models with 8- and 12-bit mantissas paired with 16-bit weight storage. We observe small accuracy improvements of $0.21\%$ and $0.43\%$ over their counterparts with narrow weight storage. We observe similar trends on other models.

We also train HBFP with various tile sizes. Tile sizes of $24 \times 24$ and $64 \times 64$ yield similar accuracy to FP32, with errors within 0.5% of the baseline. HBFP without tiles results in a larger error increase, of 0.8% over FP32, because it often forces large weight matrices to share exponents. Again, we observe similar trends on other models.

The sweet spot in the design space is HBFP with 8- to 12-bit mantissa, 16-bit weight storage and a tile size of 24. This configuration matches FP32 quality while improving arithmetic density and reducing memory bandwidth requirements. Using 8-bit mantissas reduces the memory bandwidth requirements of the forward and backward passes by up to $4\times$ compared to FP32. HBFP stores activations in floating-point format. While doing so may increase bandwidth requirements, we observe that these activations can be stored in narrow floating-point representations or even in summarized formats (e.g., for ReLU, only a single bit per value needs to be saved for the backward pass). Furthermore, activations account for a small fraction of the memory traffic when training DNNs. While activation traffic is dwarfed by weight traffic in fully connected layers, in convolutional layers the computation-to-communication ratio is so high that the memory traffic incurred by activations is not a significant throughput factor.

**HBFP vs. FP32:** Table 2 reports the validation error for all the image classification models and Table 3 reports the validation perplexity of the language modeling model. In addition, figure 3 illustrates the training process for three of the evaluated models: a WideResNet28-10 trained with CIFAR-100, a ResNet-50 trained with ImageNet, and an LSTM trained with PTB. HBFP matches the performance of FP32 in all the models and

Table 3: Perplexity of language modeling models. *hbfpX_Y* indicates an experiment with X-bit mantissas and Y-bit weight storage and a tile size of 24. All dot product operations are performed in X-bit arithmetic.

| | LSTM-PTB |
|---|---|
| fp32 | 61.31 |
| hbfp8_16 | 61.86 |
| hbfp12_16 | 61.35 |

datasets tested. We conclude that HBFP is indeed a drop-in replacement for FP32 for a wide set of tasks, leading to models that are more compact and enabling HW accelerators that use fixed point arithmetic for most of the DNNs computations.

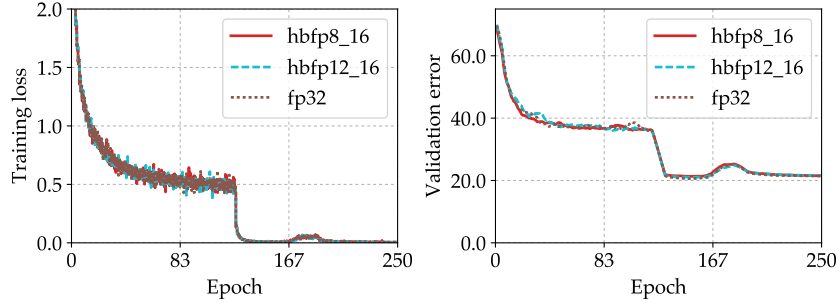

(a) WideResNet-28-10 trained on CIFAR-100 for 250 epochs.

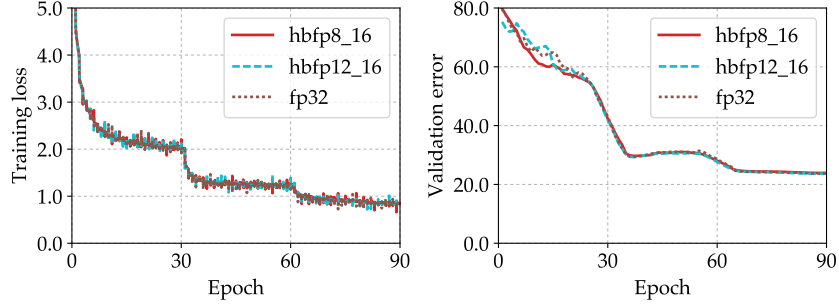

(b) ResNet-50 trained on ImageNet for 90 epochs.

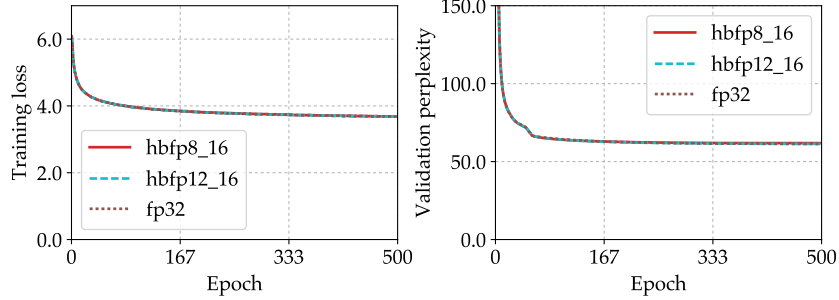

(c) LSTM trained on PTB for 500 epochs.

Figure 3: Comparison between HBFP and FP32. *hbfpX_Y* indicates an experiment with with X-bit mantissas and Y-bit weight storage and a tile size 24. All dot product operations are performed with X-bit arithmetic.

**HBFP silicon density and performance estimation:** We synthesize the accelerator on a Stratix V 5SGSD5 FPGA at a clock rate of 200MHz. We achieve a maximum throughput of 1 TOp/s using 8-bit wide multiply-and-add units in the matrix-multiplier and floating-point activations (with 8-bit mantissas plus 8-bit exponents). The activation units occupy less than $10\%$ of the FPGA resources, resulting in an $8.5\times$ throughput improvement over a variant of the accelerator that employs FP16 multiply-and-add units on the same FPGA. Finally, the conversion units occupy less than $1\%$ of the FPGA resources, and incur no performance overhead.

# 7    Conclusion

DNNs have become ubiquitous in datacenter settings, forcing operators to adopt specialized hardware to execute and train them. However, DNN training still depends on floating-point representations for convergence, severely limiting the efficiency of accelerators. In this paper, we propose HBFP, a hybrid BFP-FP number representation for DNN training. We show that the HBFP leads to efficient hardware, with the bulk of the silicon real-estate spent on efficient fixed-point logic. Finally, we evaluate HBFP, and show that, for all models evaluated, BFP-FP training matches FP32 counterparts

while resulting in $2\times$ more compact models. BFP-FP32 also leads to faster accelerators, with 8-bit BFP achieving $8.5\times$ higher throughput when compared to FP16. Higher throughput leads to faster and more energy-efficient DNN training/inference, while model compression leads to lower bandwidth requirements for off-chip memory, lower capacity requirements for on-chip memory and lower communication bandwidth requirements for distributed training.

## Acknowledgements

The authors thank the anonymous reviewers, Mark Sutherland, Siddharth Gupta, and Alexandros Daglis for their precious comments and feedback. We also thank Ryota Tomioka and Eric Chung for many inspiring conversations on low-precision DNN processing. This work has been partially funded by the ColTraIn project of the Microsoft-EPFL Joint Research Center and by SNSF grant 200021_175796.

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
