[Reviews · NeurIPS 2018]

Reviewer 1



This work suggests a new approach for fixed+floating point hybrid. To enable this, use a shared exponent per tensor that allows simple conversion between the formats. They use a fixed point representation for GEMMs to allow only mantissa multiplication while using floating point for all other operations. They also introduce a conversion scheme between intermediate representations of weights and activations. The utility of this approach is demonstrated by simulations and hardware implementation. Overall this is a nice work, with convincing arguments and experiments. Most noteworthy, I enjoyed seeing an empirical justification for favoring a large exponent, while using a trimmed mantissa. This echoes similar observations by others (e.g BF16 format used by Google). I think that additional comparison to Nervana's Flexpoint format [1] is in place, as it also used an exponent shared scheme, although with different choices for exponent and mantissa sizes. There are some aspects that I will be happy to see covered: 1) Bandwidth vs computational bound operations - your methods will probably not improve run-time for several operations, it's interesting to see some kind of tear-down for a network such as ResNet50. E.g - where were the bottlenecks for a floating-point implementation, and for yours (see [2])? 2) Batch norm – You mentioned that “While activation traffic is dwarfed by weight traffic in fully connected, in convolutional layers … not significant throughput factor”. I would argue that batch-norm is highly bandwidth bound, and that it should be discussed further in your work (most noteworthy, as batch-norm is done completely in fp32 from what I gather). 3) Estimating the cost of conversion operations – you mentioned both finding maximum value in tensor and stochastic rounding to be needed for quantization. I’ll be happy to get an estimate for the cost of these operations (runtime) and their importance in terms of accuracy. 4) A more challenging recurrent task – PTB is good, but I’ll be happy to see a more challenging benchmark such as machine translation to be assured that your methods indeed transfer to other models. Strengths: - Convincing arguments for a novel numerical representation for DNN training - Good empirical experiments Weaknesses: - Missing analysis for operations that are not computationally bound (batch-norm) - Estimating the relative cost for needed utility operations (maximum finding, stochatic rounding) [1] - https://arxiv.org/abs/1711.02213 [2] - https://arxiv.org/abs/1709.08145

Reviewer 2



This paper proposes efficient training algorithms for the block floating point (BFP) format which is often adopted for quantized computation and hardware accelerator explorations for neural network training. Specifically, the paper proposes hybrid BFP-FP training algorithm as well as approaches to manage the numerical efficiency of different mantissa and exponent bits. I find the paper very well written in the sense that, it clearly lays out the problem statement as well as existing work on this topic, highlighting the possibility of reduced bits in section 3. Section 4 then proceeds to explain the motivations of HBFP's hybrid choice, as well as approaches to manage the efficiency of BFP by tiling and storing wide weight floats. The experimentation section gives a thorough study of the different approaches as well as the baseline fp32 approaches. Overall, this is a quite solid paper that, although not mathematically deep, explores a lot of the engineering and statistics front of reduced precision training. As a result I would recommend the paper be accepted.

Reviewer 3



The authors propose to use hybrid block floating point (BFP) numbers to encode inputs/weights/activations during neural network training. BFP offers customizable dynamic range in exponents and multiplication-accumulation (MA) with BFP can be implemented with a much smaller area on a hardware chip. And similar to other approaches, authors propose to use traditional 32 bit floating point (FP32) numbers outside the MA unit, hence the name "hybrid". And empirically, hardware implementation is dominated by MA unit and the FP32 logic accounts for a very small percentage of the chip area. With GPU simulation, the authors show that HBFP can achieve comparable accuracy on a wide range of benchmarks. And the authors also describe their hardware implementation with Stratix V FPGA, which does realize the area improvement of BFP based MA unit. It is great to see such a piece of work that takes a more holistic view of hardware/software co-design to optimize system level performance of modern machine learning systems. However, upon closer inspection, I have following concerns: 1) The authors fail to clearly identify their contribution in the hybrid design. As the authors have cited in the work, there exist a large body of academia and industrial research and products in this area. It is not particularly clear to me what exactly is the innovation here. 2) The improvement to BFP is refreshing. But the authors do not really demonstrate its effectiveness. Most importantly, experiments are simply not convincing. a) In table 2, two BFP implementations use 24 and 28 bits respectively to achieve comparable accuracy to the 32 bit baseline. The saving in hardware seems to be marginal, given the change needed to adapt to the hybrid training in hardware and software. b) And for the most important FPGA implementation, the authors choose to implement a version with 8-bit mantissa and 8-bit exponent, which simply do not match any method in table 2. So we are left guessing what the accuracy would be if we took advantage of the 8.5x throughput gain from authors' claim. The experiment section definitely needs some clean up.